# Stability Assessment of p-i-n Perovskite Photovoltaic Mini-Modules Utilizing Different Top Metal Electrodes

**DOI:** 10.3390/mi12040423

**Published:** 2021-04-13

**Authors:** Janardan Dagar, Gopinath Paramasivam, Carola Klimm, Markus Fenske, Christof Schultz, Rutger Schlatmann, Bert Stegemann, Eva Unger

**Affiliations:** 1HySPRINT Innovation Lab, Helmholtz-Zentrum Berlin, Kekuléstrasse 5, 12489 Berlin, Germany; gopinath.paramasivam@helmholtz-berlin.de (G.P.); klimm@helmholtz-berlin.de (C.K.); eva.unger@helmholtz-berlin.de (E.U.); 2Young Investigator Group Hybrid Materials Formation and Scaling Kekuléstrasse 5, 12489 Berlin, Germany; markus.fenske@helmholtz-berlin.de; 3School of Engineering—Energy and Information, HTW Berlin—University of Applied Sciences, Wilhelminenhofstr. 75a, 12459 Berlin, Germany; christof.schultz@helmholtz-berlin.de (C.S.); rutger.schlatmann@HTW-Berlin.de (R.S.); Bert.Stegemann@HTW-Berlin.de (B.S.); 4PVcomB/Helmholtz-Zentrum Berlin für Materialien und Energie GmbH, Schwarzschildstr. 3, 12489 Berlin, Germany; 5Department of Chemistry & NanoLund, Lund University, Naturvetarvägen 14, 22362 Lund, Sweden

**Keywords:** self-assembled monolayer, module, electrode, perovskite, maximum power point, flakes, stability

## Abstract

Long-term stability is one of the major challenges for p-i-n type perovskite solar modules (PSMs). Here, we demonstrate the fabrication of fully laser-patterned series interconnected p-i-n perovskite mini-modules, in which either single Cu or Ag layers are compared with Cu/Au metal-bilayer top electrodes. According to the scanning electron microscopy measurements, we found that Cu or Ag top electrodes often exhibit flaking of the metal upon P3 (top contact removal) laser patterning. For Cu/Au bilayer top electrodes, metal flaking may cause intermittent short-circuits between interconnected sub-cells during operation, resulting in fluctuations in the maximum power point (MPP). Here, we demonstrate Cu/Au metal-bilayer-based PSMs with an efficiency of 18.9% on an active area of 2.2 cm^2^ under continuous 1-sun illumination. This work highlights the importance of optimizing the top-contact composition to tackle the operational stability of mini-modules, and could help to improve the feasibility of large-area module deployment for the commercialization of perovskite photovoltaics.

## 1. Introduction

Metal-halide perovskite solar cells (PSCs) have attracted attention by demonstrating a small-area power conversion efficiency (PCE) of more than 25% in the last few years [1,2,3,4,5,6,7,8]. For large-scale utilization, solar module fabrication needs to be developed with a number of interconnected cells, compared to a small-area single cell [9,10]. Series-interconnected monolithic perovskite solar modules (PSMs) can be fabricated using P1–P2–P3 patterning steps [2,11,12], as in any other thin-film solar technology like CdTe or CIGS. The deposition and patterning steps are as follows. Initially, subcells are defined by patterning the ITO bottom electrode with a P1 scribe, then all active layers of the solar cell are deposited, and are then separated from each other by the P2 patterning step [9,13]. Subsequently, the metal back electrode is deposited and a final, P3 scribe separates the metal top contacts of the sub-cells. The principles of laser patterning of thin film have been widely discussed in fabrication of modules [14,15,16]. In p-i-n PSMs, p and n are hole and electron transport layers, and “i” refers to the perovskite layer [2]. As a preference, p-i-n PSMs can be fabricated using easy and low-temperature processing steps. In addition, this includes less absorptive contact layers and shows low current density–voltage hysteresis [17]. However, several challenges have been addressed in the processing of highly efficient and stable p-i-n, such as poor wettability of the hole transport layer, upscaling of the perovskite layer and long-term stability of the perovskite modules [18,19]. In our previous work, we have extensively studied both picosecond and nanosecond laser pulses as used for P2 [9] and P3 [16] patterning steps to fabricate the mini-modules. We optimized the shorter picosecond laser pulses to remove the PbI_2_ debris, generated during P2 and P3 patterning, and improved the performance of the perovskite mini-module [9,16].

With regard to the P3 patterning step, it has been reported that flaking, or the formation of large metal flakes or particles of top metal electrode, causes problems with the electrical isolation of the series interconnected sub-cells in modules [20]. The flaking causes fluctuations in the maximum power point (MPP) tracking under continuous 1-sun illumination, which result in a reduction in module performance and long-term stability due to intermittent lack of contact with the sub-cells [21]. To decrease the possibility of flaking of the metal electrode, several strategies have been applied, such as optimization of nanosecond or picosecond laser pulse, and wider P3 scribing with multiple time overlapping [16]. In addition, laser patterning with N_2_ gas flow is used to remove the metal particles [21]. However, these solutions have been found difficult to employ for Cu and Ag metal top electrodes. Rather than being blasted away during the laser-ablation process, the metal seems to locally melt or rupture, creating metal flakes that point upward from the laser-scribe but remain physically connected to the metal back electrode. Consequently, it is highly important to resolve the issue of flaking back/top electrodes to make highly efficient and stable perovskite solar mini-modules.

In this study, we demonstrate a solution to this issue by using a bilayer metal electrode, via depositing a thin 30 nm-thick layer of additional gold (Au) on top of the 70 nm-thick copper (Cu) electrode by thermal evaporation [22,23]. The PSMs of three series interconnected cells with a Cu/Au bilayer top electrode exhibited an improved steady-state efficiency of 18.9% after continuous 11 h 1-sun illumination. In addition to the single-metal electrode, some alloyed or composite metal electrodes have also been investigated previously, in perovskite solar cells such as AgAl (silver–aluminum) [24] and NiAu (nickel–gold) [25]. However, the introduction of double metal layers (CuAu) has not been used to fabricate the p-i-n perovskite mini-module, because of the concern of long-term stability assessment. The PSMs with a Cu/Au electrode nullified the fluctuations which occurred in the MPP due to the formation of Cu metal particles at different interconnections of the cells. This module strategy enables the development of a highly efficient and stable large-area module.

## 2. Results and Discussion

In this work, perovskite solar mini-modules with a p-i-n configuration of Glass|ITO|2PACz|Cs_0.15_FA_0.85_PbI_2.55_Br_0.45_|C_60_|SnO_2_|Cu|Au were fabricated as shown in Figure 1a. A detailed explanation of the PSM fabrication process is given in the experimental section. In short, PSMs were built on conductive indium tin oxide (ITO) substrates, prepared by P1 patterning. At first, high-quality hole-selective contacts with a self-assembled monolayer (SAM) of (2-(9H-carbazol-9-yl) ethyl) phosphonic acid (2PACz) were spin-coated on glass/ITO substrates. As a photoabsorber, CsFA (Cs_0.15_FA_0.85_PbI_2.55_Br_0.45_) was deposited over the 2PACz layer. Next, a layer of 23 nm of C_60_ was thermally deposited over the perovskite layer, which is further followed by ALD (Atomic Layer Deposition), which deposited a 20 nm SnO_2_ layer, and the P2 patterning step was performed. Here we introduced three different top metal electrodes, such as Ag, Cu, and Cu covered with Au (Cu/Au) via thermal evaporation, and P3 patterning was performed to cut the metal electrode and connect subcells in series interconnection. Figure 1b showed the current flow in interconnected subcells to the mini-module. The layout of the 2.20 cm^2^ module has been also shown in Figure 1c. The geometrical fill factor was calculated to be 91%. The aperture area used during characterization of the mini-module was 2.35 cm^2^, resulting in an aperture efficiency of 16.60%. The non-active area of 0.15 cm^2^ includes area losses due to the P1–P2–P3 interconnection.

Figure 1d shows the steady-state maximum power point efficiencies for the best-performing modules of each type over more than 11 h under continuous 1-sun illumination at 25 °C. Interestingly, we found that the long-term efficiency for Cu-based PSMs fluctuates with time, which further results in a complete breakdown of the PCE of the modules. The fluctuation in PCE is due to the fluctuation of steady-state current (I_MPP_), while the steady-state voltage (V_MPP_) was stable over time, as shown in Appendix A. Similar fluctuations in steady-state efficiency were also observed with a Ag electrode-based PSM, but were comparatively higher than those of a Cu-based PSM. In this study, we modified metal top-electrodes by depositing an additional 30 nm-thick gold (Au) layer on top of the 70 nm thick copper (Cu) electrode (Cu/Au), which improved the photovoltaic performance and removed the fluctuations in steady-state efficiency of the PSMs, as shown in Appendix A.

The current-voltage (*I*–*V*) curves of the best-performing 2PACz-based PSMs fabricated using different top electrodes are shown in Appendix A. The photovoltaic (PV) parameters of PSMs, such as short circuit current (I_SC_), open-circuit voltage (V_OC_), fill factor (FF), power conversion efficiency (PCE) obtained from IV curves, and power conversion efficiency at maximum power point (PCE_MPP_) are summarized in Table 1. The PSMs with a Cu electrode exhibited superior performance compared to Ag-based PSMs. In particular, a Cu/Au-based PSM yields a remarkable efficiency of 18.3%, as shown in Table 1. *I*–*V* curves of PSMs were also measured under dark conditions, as shown in Appendix A. We observed that PSMs with Cu/Au showed a low leakage of current compared to those with Ag- and Cu-based PSMs. We have also calculated the ideality factors for all PSMs with different top electrodes from their dark *I*–*V* curves, showing that Cu/Au exhibits a lower ideality factor of 0.288 compared to the Cu-based PSM 1.02 (as shown in Appendix A). This demonstrates that Cu/Au PSMs lead to better charge extraction, and thus higher FF values. As shown in Appendix A, MPPs have also been measured for a short time, showing a maximum PCE of 18.9% compared to Ag- and Cu-based PSMs.

Figure 2 shows the cross-section scanning electron microscopy (SEM) images ((Figure 2a–c) tilted by 10°) of PSMs, based on Figure 2a,d for Ag, Figure 2b,e for Cu and Figure 2c,f for Cu/Au top electrodes. We observed that Ag and Cu metal top electrode-based PSMs display Ag and Cu flakes (conductive metal particles), respectively, at the interconnection of the cells of modules. The isolation of the P1 or P3 scribe line caused redeposition of conducting particles after removal, as shown in Figure 2a,b,d,e. These flakes are responsible for the steady-state efficiency fluctuation, which further results in cell exclusion from the module circuit while stressing the module for a long period under 1-sun illumination at 25 °C. Optical microscopy images of the P2 and P3 patterning of the perovskite mini-module are also shown in Appendix A with different electrodes. The formation of these conductive particles can be avoided by using an extra layer (30 nm) of gold (Au) over the copper (Cu) (70 nm) metal electrode (Cu/Au) [21]. Gold has the capability to bind strongly to the bottom layer and thus protects the Cu layer from forming the flakes, due to its high degree of thermal expansion [26,27]. EDX images of P3 scribing of both Cu- and Cu/Au-based PSMs are shown in Appendix A.

The most reliable metric to compare the module stability to the integrated lifetime energy yield (LEY) is calculated using the following equation [28,29]:$$\mathrm{LEY}\;=\;\int_{0}^{t}PCE\;\left(t\right)dt$$
where “*t*” refers to time. Figure 3 shows the LEY calculations for PSMs with different electrodes. The LEY for Cu/Au-based metal contacts showed that these modules had the highest figures, compared to the LEY produced by only Cu- and Ag-based perovskite mini-modules. This indicates that the modules showing the higher LEY values lead them to become more relevant for testing according to international standards such as the International Electrotechnical Commission (IEC) 61215 standard [30,31].

## 3. Conclusions

In summary, fully laser-patterned monolithic p-i-n perovskite solar mini-modules have been fabricated comparing different metal electrodes as back contacts, namely, Cu, Ag or Cu/Au bilayers. We find that using Cu or Ag only leads to metal-back contact delamination and flaking upon P3-laser scribing, while the Cu/Au bilayer exhibits a very clean cut with good adhesion remaining between the device and metal contact. The perovskite modules with a Cu/Au bilayer top electrode therefore outperform devices with Ag or Cu metal contacts, exhibiting close to 19% steady-state efficiency with negligible loss >11 h under continuous simulated AM1.5G illumination in an inert atmosphere.

## 4. Experimental Section

### 4.1. Materials

SAM (2PACz), lead (II) iodide (99.99% trace metals basis), and lead bromide (PbBr_2_) were purchased from Tokyo Chemical Industry (TCI). Formamidinium iodide (FAI) and Cesium iodide (CsI) were obtained from Dynamo GmbH (Dresden, Germany) and abcr GmbH (Karlsruhe, Germany). Ethanol (anhydrous) was achieved from VWR Chemicals. Anisole, DMF, and DMSO chemicals were purchased from Sigma-Aldrich (St. Louis, MO, USA). All the chemicals were used as supplied without any further purification.

### 4.2. Perovskite Precursor Inks Preparation

The 1.2 M concentration of CsFA (Cs_0.15_FA_0.85_PbI_2.55_Br_0.45_) perovskite ink was prepared using the following procedure. Initially, PbI_2_ (446.4 mg), PbBr2 (121.6 mg), FAI (185.5 mg), and CsI (57.4 mg) powders were scaled and mixed in vials, and dissolved with 0.750 mL of DMF and 0.250 mL of DMSO. The vial is kept on a shaker at 60 °C for 2 h. The resulting perovskite ink was filtered using a 0.2 µm-sized polytetra-fluorethylene (PTFE) filter.

### 4.3. Perovskite Mini-Module Fabrication

We fabricated a perovskite mini-module with an inverted (p-i-n) planar structure, with a layer configuration of ITO|2PACz|Cs_0.15_FA_0.85_PbI_2.55_Br_0.45_|C_60_|SnO_2_|CuAu, or Ag or Cu, using the following procedure. The perovskite mini-module fabrication is achieved by laser scribing of different layers using a Rofin-Baasel laser-patterning tool, see [9] for more details. In the P1 scribing process, indium tin oxide (ITO) glass substrates (of 25 mm × 25 mm, resistivity of 15 Ω sq^−1^) were subjected to a 1064 nm, picosecond (ps) laser (500 kHz, POD2, fluence of 7.33 J/cm^2^, speed of 400 mm/s) to cut the ITO electrode [9,16]. The patterned ITO substrates were cleaned sequentially according to the same procedure as described previously [2]. After that all substrates were transferred in a N_2_-filled glove box, where 1.2 mmol/L of 2PACz (MW = 275.24) solution was deposited on cleaned ITO substrate via spin-coating at a speed of 3000 rpm for 30 s. The substrates were immediately kept on a hotplate at 100 °C for 10 min. The CsFA perovskite layer was deposited on ITO/SAM substrates via spin-coating, with a spin speed of 3500 rpm with 5 s acceleration until 35 s steady duration. Anisole was used as an antisolvent. When 10 s were remaining to finish the spin coating, 250 µL of anti-solvent was dropped on the wetted perovskite layer. Next, the substrates were transferred to a hotplate at 100 °C for 30 min in a N2 atmosphere.

All the substrates were transferred in the evaporation chamber, where C_60_ (23 nm) was thermally evaporated on top of the perovskite layer, at a rate of 0.05–0.12 A°/s with a base pressure of 1 × 10^−6^ mbar. Next, 20 nm of a SnO_2_ layer was deposited using an atomic layer deposition (ALD) technique, using an Arradiance GEMStar reactor. For the 20 nm deposition of the SnO_2_ layer, 140 cycles were applied at an 80 °C substrate temperature. The layer stack of ITO|2PACz|Cs_0.15_FA_0.85_PbI_2.55_Br_0.45_|C_60_|SnO_2_ was patterned through a 532 nm ps laser (10 ps, pulsed at 100 kHz, POD5 and speed 200 mm/s, a fluence of ~1.49 J/cm² with 3 passes) which is referred to as P2 scribing. For the top electrodes, Cu or Ag electrodes of 100 nm thickness were thermally deposited. For Cu/Au electrode deposition, at first, 70 nm of Cu was deposited, and immediately after, 30 nm gold was evaporated over the Cu in a metal evaporator at a base pressure of 1 × 10^−6^ mbar. P3 patterning was carried out to cut different top metal electrodes, to connect 3 cells in series using a ps laser at 532 nm (100 kHz, POD5, speed of 500 mm/s, three times scribing, each time with a fluence of 3.04 J/cm^2^). The active area for a single cell is calculated to be 0.734 cm^2^, and for the module, 2.2 cm^2^, which is determined by microscopic imaging.

### 4.4. Sample Characterization

#### Solar Mini-Module Characterization

Current density–voltage (*J*–*V*) measurements of PSMs were performed, under AM 1.5 G 1000 Wm^−2^ under STC conditions using a sun simulator of AAA class calibrated with a silicon reference cell (Fraunhofer ISE, Freiburg, Germany), according to the same procedure as described previously [2].

### 4.5. SEM Measurements

SEM images were captured from a Hitachi S-4100 at 5 kV acceleration voltage system using a Zeiss Merlin.

## Figures and Tables

**Figure 1 micromachines-12-00423-f001:**
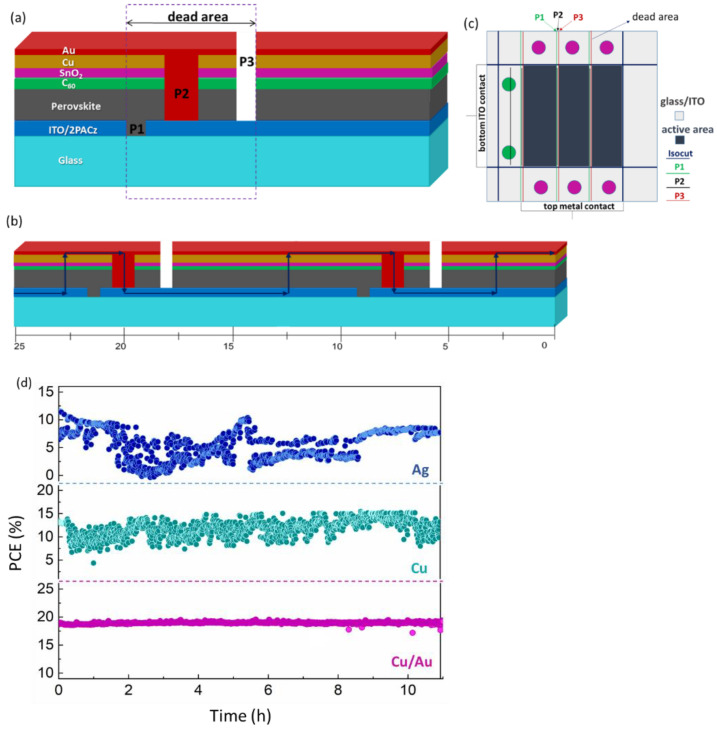
(**a**) The schematic device layout of p-i-n perovskite solar mini-modules with P1, P2 and P3 interconnections; (**b**) charge flow in the series connection among cells in a module; (**c**) layout of a 2.20 cm^2^ module; and (**d**) long-term maximum power point tracking (MPPT) of p-i-n type perovskite solar modules (PSMs) with different metal top electrodes, such as Ag, Cu and Cu/Au, while measured under simulated 1-sun illumination.

**Figure 2 micromachines-12-00423-f002:**
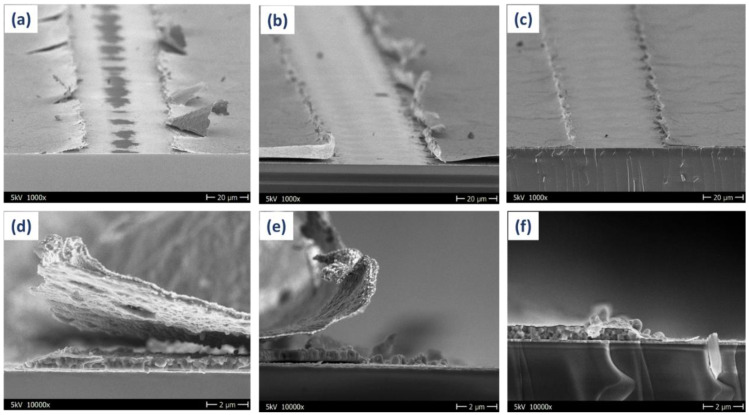
Scanning electron microscopic images of P2 and P3 patterning of (**a**,**d**) Ag electrode, (**b**,**e**) Cu electrode and (**c**,**f**) Cu/Au electrode of a perovskite mini-module.

**Figure 3 micromachines-12-00423-f003:**
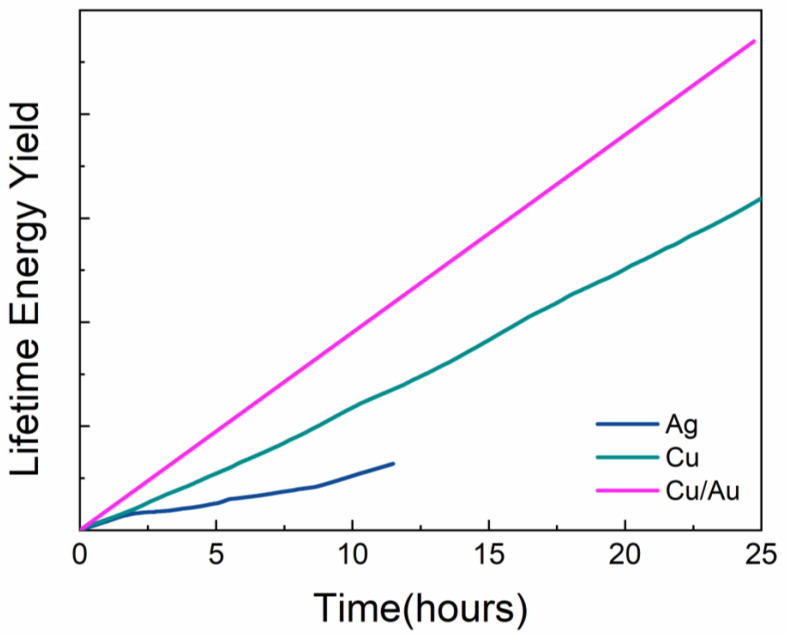
Integrated lifetime energy yield (LEY) of perovskite solar mini-module (PSM) with different top electrodes (Ag, Cu and Cu/Au).

**Table 1 micromachines-12-00423-t001:** Summarized photovoltaic (PV) parameters of perovskite solar mini-modules with different metal top electrodes such as Ag, Cu and Cu/Au, while measured under simulated 1-sun illumination.

PV Parameters	Ag	Cu	Cu/Au
I_SC_ (mA)	16.27	15.93	15.83
V_OC_ (V)	3.36	3.39	3.40
FF (%)	68.3	71.7	74.1
PCE (%)	17.0	17.6	18.3
PCE_MPP_ (%)	16.9	17.3	18.9

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
