# Peer review of "Stability Assessment of p-i-n Perovskite Photovoltaic Mini-Modules Utilizing Different Top Metal Electrodes"

_micromachines, 2021, doi:10.3390/mi12040423_

Round 1

Reviewer 1 Report

The authors of the manuscript we demonstrated Cu/Au metal-bilayer based p-i-n type perovskite solar modules (PSMs) with an efficiency of 18.9% on an active area of 2.2 cm2 under continuous 1 sun illumination. The work presented in the manuscript is of scientific interest and suitable for the readers of the journal. However, the authors may be advised to update their manuscript according to the following comments.

  1. Line # 40: Reference 7 is 3 years old and not relvent to the corresponding statement.
  2. Line #74 & 182: The authors mentioned that p-i-n configuration of Glass|ITO|2PACz| Cs0.15 FA0.85PbI2.55Br0.45|C60|SnO2|Cu|Au were fabricated as shown in Figure 1a but Figure 1a does not show SnO2 layer.
  3. Line #167: Writing this “Self-assembled monolayer (2PACz)” may confuse the reader since the author previously mentioned Self-assembled monolayer (SAM). It is better to write “SAM of 2PACz”.
  4. MPP has been defined only in the abstract not in the main text.
  5. Efficiency of a solar cell is measured from maximum power point (MPP). So writing “long-term MPP efficiency” is redundantThe authors may write “long-term efficiency” since they mentioned PCE (%) in the Figure 1d.
  6. Author did not mention the units of the photovoltaic parameters in the Table 1.
  7. Isc, Voc, FF, and PCEMPP were not defined in the manuscript.
  8. What is the difference between PCE and PCEMPP?
  9. Caption of Figure S3 is not correct. The authors mentioned “Current (mA)” in the label for y-axis but in caption it is “Current density”. Also, the author should update the statement in Line # 110 accordingly.
  10. The authors did not provide performance data of the unit cells.

Author Response

uploaded

Reviewer 2 Report

The reviewer suggests revision:

  • The paper presents the stability assessment of p-i-n perovskite photovoltaic mini- modules utilizing different top metal electrodes but the authors should highlight: what are the original achievements of the presented work, what is the novelty compare to the standard produced electrodes. Did anyone use of bilayer metal electrode in perovskite photovoltaic mini- modules. If so, what kinds of materials are used in this type of solar cells? It should be included in the literature review.
  • English should be improved.
  • It is difficult to evaluate some of the results because there are no figures: S3, S4, S5, S6, S7, S8 in the presented paper.
  • Some sentences in the manuscript are not clear.
  • Remove the repetition of words in sentences e.g.: line 143“compare”, line 167-172 “purchased” (6x), line 177 “and” (3x).
  • There is no explanation of the abbreviations e.g.: BCP (Fig.1), CsFA (line 175).
  • In Table 1 the units are omitted.
  • There are some editorial errors e.g.: line 123: …”Ag. Cu and Cu/Au…”- dot
  • These are not all the comments. Authors should specifically check and correct the manuscript!!!

Round 2

Reviewer 1 Report

The author has thoroughly addressed all my comments. I appreciate the effort the authors have put into addressing my comments and I am now fully satisfied that the manuscript can be considered for publication in its present form.